# COVID-19 Vaccine Hesitancy and Related Factors among Unvaccinated Pregnant Women during the Pandemic Period in Turkey

**DOI:** 10.3390/vaccines11010132

**Published:** 2023-01-05

**Authors:** Mehmet Akif Sezerol, Selin Davun

**Affiliations:** 1Doctorate Program, Institute of Health Sciences Epidemiology, Medipol University, Istanbul 34810, Turkey; 2Health Management Program, Graduate Education Institute, Maltepe University, Istanbul 34857, Turkey; 3County Health Department, Sultanbeyli, Istanbul 34935, Turkey

**Keywords:** COVID-19, vaccine, pregnancy, hesitancy, acceptance

## Abstract

The COVID-19 virus appeared in Wuhan, China in 2019 and spread rapidly all over the world. Vaccination guidelines have recommended pregnant women to get vaccinated against COVID-19 to prevent disease. This study aimed to understand the willingness of pregnant women to vaccinate and the factors associated with their hesitation and resistance. This cross-sectional study was conducted between March and April 2022. The sample size was not calculated and tried to reach all pregnant women who did not have any COVID-19 vaccine. In the first part of the questionnaire, socio-demographic questions, and in the last part, the short form of the 12-item anti-vaccine scale, which was prepared as a 5-point Likert scale, were applied. The study was completed with 561 pregnant women. The mean score of the pregnant women who participated in this study from the vaccine hesitancy scale was found to be 33.40 ± 6.07. It was found that as the income of pregnant women decreased, the vaccine hesitancy scores decreased. Vaccine hesitancy scores of those who stated no one gave advice were significantly higher. The vaccination of pregnant women will bring significant success to the fight against the COVID-19 pandemic when implemented as part of the public health policies of countries.

## 1. Introduction

The COVID-19 pandemic, which emerged in December 2019 and continues today, affects the whole world and causes an increasing number of deaths daily. Pregnancy is a process that makes women vulnerable to viral infections and causes partial suppression of the immune system. In Iceland, during the influenza epidemic, mortality rates of 1918 cases were reported to be 2–6% for the entire population and 37% for pregnant women. Considering this situation, it can be said that pregnant women are in the risk group regarding COVID-19 infection [1]. Due to the sensitivity of pregnant women to severe pneumonia, the COVID-19 infection process should be continued in a more controlled manner. Therefore, pregnant women constitute a more sensitive population in infection prevention strategies [2]. In a systematic review of 192 studies involving more than 64,000 pregnant women with a suspected or definitive diagnosis of COVID-19, it was reported that: 17.4% of the patients had pneumonia, 17.1% received oxygen support, 13.4% had acute respiratory distress syndrome (ARDS), 11.3% had severe disease, 3.3% had intensive care, 1.6% received invasive ventilation, 0.11% received extracorporeal membrane oxygenation (ECMO), and death was observed at a rate of 0.8% [3].

Vaccination is one of the biggest contributors to global health in world history and plays a major role in protecting public health [4,5]. Serious measures have been taken worldwide against the spread of COVID-19 and vaccine studies have been started in many different countries in a short time [6]. Vaccination during pregnancy should be preferred as it reduces the risk of infection and protects the mother and baby [7]. The Centers for Disease Control and Prevention (CDC) and the World Health Organization (WHO) strongly recommend the vaccination of pregnant and lactating people [8]. In addition, the American Society of Gynecology and Obstetrics (ACOG) states that pregnant women should be encouraged by healthcare professionals to talk about their vaccination plans, who can then address their concerns [9]. A study suggests that the COVID-19 vaccine administered during pregnancy may transmit antibodies to fetuses or newborns through the placenta and breast milk, thus providing immunity [10]. In light of all these recommendations and studies, it is recommended to vaccinate pregnant and lactating mothers for better maternal and fetal outcomes during the pandemic period [11].

Deficiencies in studies on the efficacy and safety of current vaccines in pregnant women lead to hesitations about administering the COVID-19 vaccine during pregnancy and lactation [12]. Providing pregnant women with timely and accurate data on the COVID-19 vaccine is important in increasing vaccine confidence [7]. Although it has been demonstrated by studies that vaccines protect against death and disability, vaccine indecision and opposition, which are increasing and negatively affecting public health, continue as a major public health problem. Despite the evidence supporting the safety and efficacy of the COVID-19 vaccine in pregnant women and the recommendations made for the vaccination of pregnant women, hesitations about the COVID-19 vaccine in pregnancy are significant.

This study aimed to evaluate the vaccination hesitations of pregnant women who refused to be vaccinated against COVID-19 during the pandemic period and the factors that may affect these decisions.

## 2. Materials and Methods

### 2.1. Type of Research

This study is a cross-sectional study.

### 2.2. Study Population

This cross-sectional study was conducted between March and April 2022 in 1277 pregnant women who had never been vaccinated against COVID-19 in the records of the District Health Directorate in Sultanbeyli district of Istanbul. The population of Sultanbeyli district is 349,485 and 48.8% of them are women. Another feature of Sultanbeyli district is that it is the lowest district of Istanbul in terms of the socio-economic development index [13]. The number of active pregnant women is 3492 in March 2022 in the district, where approximately 6000 deliveries occur annually. The total number of pregnant women who are unvaccinated by COVID-19 was 1277 in March. The sample size was not calculated in the study; we tried reach all of the pregnant women who did not have any COVID-19 vaccine. 59 of the participants could not be reached because they did not have a registered phone number in the system or because the wrong number was registered. 139 of unvaccinated pregnant women had foreign nationalities which could not be included in the study because of the language barrier. In addition, 367 of them could not be reached because they did not answer the phone. Finally, 151 of unvaccinated pregnant women refused to participate in the study. Therefore, the study was completed with 561 pregnant women. The selection of the participants is shown in Figure 1 in detail.

### 2.3. Measuring Tools

A questionnaire consisting of five parts was applied to the pregnant women via telephone. Before the application of the questionnaire, training was given to the people who would make phone calls to pregnant women. In the first part of the questionnaire, socio-demographic questions such as age, the month of pregnancy, education level, employment status, income level, the effect of the pandemic period on income, and the number of children were included. Income status of individuals was determined on the basis of 4682 TL, which is the hunger limit of a single person in Turkey at the time of data collection. Accordingly, the income status of the participants was divided into two groups, below and above the limit of 4682 TL. In the second part, the situation of having COVID-19 infection, the history of hospitalization, and the risk of contracting COVID-19 were asked. In the third part, questions were asked about perceptions of health level, place of pregnancy follow-up, previous vaccination history, vaccination status of the child, and the desire to have tetanus vaccination during pregnancy. In the fourth section, it was asked who made these recommendations and the situation of receiving advice on whether or not to have the COVID-19 vaccine. In the last part, the short form of the 12-item anti-vaccine scale prepared by Kılınçarslan et al., which was prepared as a 5-point Likert scale, was applied. The Cronbach’s alpha value for the short form was found to be 0.855. In the short form, there are 4 items about the benefit and protective value of the vaccine, 5 items about anti-vaccination, and 3 items about the solutions for not being vaccinated. The scale does not have a cut-off value, and as the score increases, vaccine hesitancy increases [14].

### 2.4. Statistical Analysis

Statistical Package for the Social Sciences (SPSS) Program version 22.0 trial version was used for statistical analysis. Continuous variables were expressed as mean ± standard deviation (SD). Categorical variables were expressed as numbers and percentages (%). Chi-square and Fisher’s exact tests were used to compare the categorical variables between groups. For statistical analysis of the data, the Mann–Whitney U test and Kruskal–Wallis analysis of variance were used to compare continuous variables that did not fit the normal distribution. The conformity of the variables to the normal distribution was examined using visual (histogram) and analytical methods (Kolmogorov–Smirnov/Shapiro–Wilk). Logistic regression analysis was performed by dividing the participants into two groups, those below the average and those above the average according to their average scores. Variables that were found to be significant in univariate analyses were included in the logistic regression analysis. A *p*-value below 0.05 was considered significant statistically.

### 2.5. Ethical Considerations

Prior to starting this study, Ethics Committee Approval and research permits were obtained from the Medipol University Ethics Committee with 341 protocol number, and the people who constituted the sample size of the research were asked to participate in the study after being informed about the research and permits. Our study was conducted according to the Declaration of Helsinki and written informed consent was obtained from all participants.

## 3. Results

This study was completed with 561 pregnant women. The mean age of the pregnant women participating in the study was 28.07 ± 5.03. 54.4% of the pregnant women are primary school graduates, 25.7% are high school graduates, 31.4% have no children, and 68.6% have at least one child. More than half of the participants (53.5%) have a monthly income of less than 4682 TL. Figure 2 shows the distribution of reasons for refusal of pregnant women who refused to be vaccinated against COVID-19. While answers were given such as scientific reasons, conspiracy theories, not trusting the health authorities, not believing in the effect of the vaccine, having had a previous COVID-19 infection, 62.7% stated that they were afraid of the possible side effects of the vaccine and therefore did not want to be vaccinated against COVID-19. 

The mean score of the pregnant women who participated in this study from the vaccine hesitancy scale was found to be 33.40 ± 6.07. It was found that as the income of the pregnant women decreased, the vaccine hesitancy scores decreased (*p* < 0.002). 89.4% of pregnant women who have at least one child stated that their children have had their childhood vaccinations up to now. No significant correlation was found between the occupational status of the pregnant women and their vaccine hesitancy scores (*p* > 0.005). The distribution of sociodemographic characteristics of pregnant women and their relationship with vaccine hesitancy scores are shown in Table 1 in detail.

When questioned whether the pandemic had any effect on the income of the participants, 74.2% stated that it did, and the vaccine hesitancy scores of those who stated that it did not affect it were found to be significantly higher (*p* = 0.013). It was also questioned whether the pregnant women had tetanus vaccination according to the vaccination program in Turkey. Although 83.8%, a majority, of them stated that they had the tetanus vaccine, the vaccine hesitancy score was found to be significantly higher in those who did not (*p* < 0.001). In this study, no significant relationship was found between the cases of being infected with COVID-19 or having a history of hospitalization, and vaccination hesitancy scores (*p* > 0.005).

When the pregnant women who participated in this study were asked whether they received any advice from anyone about getting vaccinated, 59% stated that they received advice. Vaccine hesitancy scores of those who stated that no one gave advice were also found to be significantly higher (*p* = 0.015). When the pregnant women who received advice from anyone were asked from whom they received advice, 24.2% of the pregnant women stated that they received advice from their family doctor and only 18.5% from a specialist physician. Vaccination hesitancy scores of those who were advised by their family physicians to be vaccinated were found to be significantly lower (*p* = 0.013). No significant relationship was found with whether or not they received advice from specialist physicians (*p* > 0.05). There was no significant relationship between whether pregnant women received advice from other people (relatives, spouses, friends, others) and their hesitations about getting vaccinated (*p* > 0.05) (Table 2).

Vaccine hesitancy scores of the participants were divided into two parts, below and above the mean, and results that were found to be significant in univariate analyses such as education status, chronic disease, income, the effect of the pandemic on income, whether someone recommends the COVID-19 vaccine, and the status of having tetanus vaccine were included in the logistic regression analysis. In the results of multivariate analysis, it was determined that those who had the tetanus vaccine constituted the reference group, and those who did not have vaccine hesitancy were 3.6 times more (*p* < 0.05). It was determined that those whose income status is above 4682 TL showed a significant 1.5 times more vaccination hesitancy. The analysis results are shown in Table 3 in detail.

## 4. Discussion

Vaccination hesitancy is important in the whole population, and it is more important in populations with risk groups. In pregnant women who are in the risk group, this situation requires extra attention and awareness as it can have an impact on both their health and the health of newborns. This study has been one of the studies that shed light on this awareness by working both with pregnant women who are in a risk group and with those who have not been vaccinated against COVID-19.

In this study, when the educational status and vaccination hesitancy scores of the participants were examined, it was found that as the education level increased, the scores obtained from the vaccine hesitancy increased significantly and the group with the highest vaccination hesitancy was university graduates. Another study conducted in France found that participants with higher education were statistically more vaccinated and more associated with vaccine acceptance [15]. In a study conducted in Japan, people with less than 13 years of education were found to be significantly more hesitant about vaccination [16]. In this study, people with higher education levels were more hesitant about vaccination, which may be due to the ease of access to social media and misinformation. When the pregnant women in this study were questioned whether they had a chronic disease or not, 91% of them stated that they did not have any chronic disease, but the vaccine hesitancy scores of those with chronic diseases were significantly lower. In a population-based study conducted in Hong Kong, similar to this study, it was found that those with chronic diseases showed significantly less vaccine hesitancy [17]. Again, in a study conducted in France on COVID-19 vaccine hesitancy, it was found that those with chronic diseases stated fewer hesitations about vaccination than those without [18]. When the income status of the participants in the study was examined, a significant difference was found again, and those with higher incomes showed higher vaccine hesitancy scores. In a study conducted with pregnant women in Romania, it was found that, unlike this study, those with a monthly income level above the average were 1.13 times more hesitant, and those with a lower income 2.52 times more [19]. The differences in our findings at the income level from other studies may be that people with higher incomes have easier access to more sources that may be effective in spreading misinformation on social media, and other misinformation. When the participants were asked whether they had received any advice from a healthcare professional, spouse, relative, or friend about getting vaccinated, 59% stated that they received a recommendation, although this rate is higher than those who did not. Despite the sources suggesting that COVID-19 vaccines are safe for pregnant women, it is a problem that not all health personnel encourage pregnant women to be vaccinated. In this study, it was observed that the pregnant women who were recommended to be vaccinated by health personnel had significantly lower vaccination hesitancy scores, showing how important the role of health personnel is in the vaccination acceptance of pregnant women.

The reasons for not vaccinating pregnant women who were not vaccinated against COVID-19 in this study were also examined. It was stated by the participants that fear of the side effects of the vaccine was the most common cause. Then, reasons such as thinking that the vaccine is ineffective, thinking it is unnecessary, and not finding enough time to be vaccinated are listed. Reasons such as thinking that there is a conspiracy theory, not believing in the pandemic, scientific reasons, and religious beliefs have been put forward to a small extent. In a study conducted on Twitter in Turkey, in which 1021 tweets about the COVID-19 vaccine were evaluated, it was determined that insufficient scientific evidence was mentioned the most, and conspiracy theory was mentioned secondly [20]. Again, in a study evaluating patient factors affecting COVID-19 vaccination in pregnant women, the rate of not trusting the COVID-19 vaccine was found to be significantly higher in unvaccinated pregnant women [21]. Similar to this study, Berry et al.’s study in the United States also raises concerns that the vaccine is being developed too quickly and has not undergone adequate scientific processes [22]. In a study conducted to examine the acceptance of COVID-19 vaccines by society in Australia, it was determined that trust in the health system and government policies, which was also stated in this study, is positively related to vaccine acceptance, and the more the trust increases, the more the vaccine is accepted [23]. In this context, it is understood that the increased trust in health authorities will positively affect the acceptance of COVID-19 vaccines as well as other types of vaccines. The reason for the higher rate of rejection due to side effects compared to other reasons in this study may be the low socioeconomic level of the region studied and the difficulties in accessing scientific resources.

This study is a strong study with several samples and is to be conducted on pregnant women who do not have the COVID-19 vaccine. On the other hand, the fact that the sample of the study was only unvaccinated pregnant women may cause a selection bias as the mean score of vaccine hesitation is higher than that of the normal population. Important findings were obtained in the study, and it is an important study in revealing the main reasons underlying the hesitations of pregnant women who are hesitant about vaccination and will guide the political and academic studies to be carried out. There are also some limitations. First of all, although it was aimed to reach the entire unvaccinated pregnant population in the region, only 43.93% of them could be reached, as stated in the method section; this resulted in lower participation than expected due to the large number of pregnant women who did not respond and could not be reached. Secondly, due to the absence of a cut-off point in the vaccine hesitation scale, individuals could not be classified as definitely having or not having vaccine hesitancy, and they were evaluated according to their scores. Another limitation of this study is that it was conducted only on pregnant women in Sultanbeyli. Since the Sultanbeyli region of Istanbul has the lowest socioeconomic level, it should be considered an essential factor in vaccination hesitancy and not being vaccinated by pregnant women. In order to prevent this limitation, multicenter studies should be conducted in the future. Finally, there is the possible existence of other unexamined but potentially influential variables.

## 5. Conclusions

Pregnant women constitute a high-risk group for severe complications of infection during the COVID-19 pandemic. It is also essential for pregnant women to have the COVID-19 vaccine to protect the health of themselves, their babies, the people around pregnant women, and the healthcare professionals who will provide healthcare services. Studies have shown that COVID-19 vaccine applications reduce maternal and perinatal mortality and morbidity, and pregnant women are supported in be vaccinated. Pregnant women’s hesitations about the COVID-19 vaccine may impact their future approach to vaccines that they can get their children to have and may have more significant public health consequences. Raising awareness of vaccinations and their benefits should be done by health personnel with pregnant women according to available resources. Information on vaccine applications and effects in pregnancy should be presented to pregnant women and those who may have an impact on them, understanding their concerns, without judgment, with an evidence-based and sensitive approach. It will be more effective if the people who provide this information are health professionals with updated and accurate information. The vaccination of pregnant women will bring significant success in the fight against the COVID-19 pandemic when implemented as part of the public health policies of countries.

## Figures and Tables

**Figure 1 vaccines-11-00132-f001:**
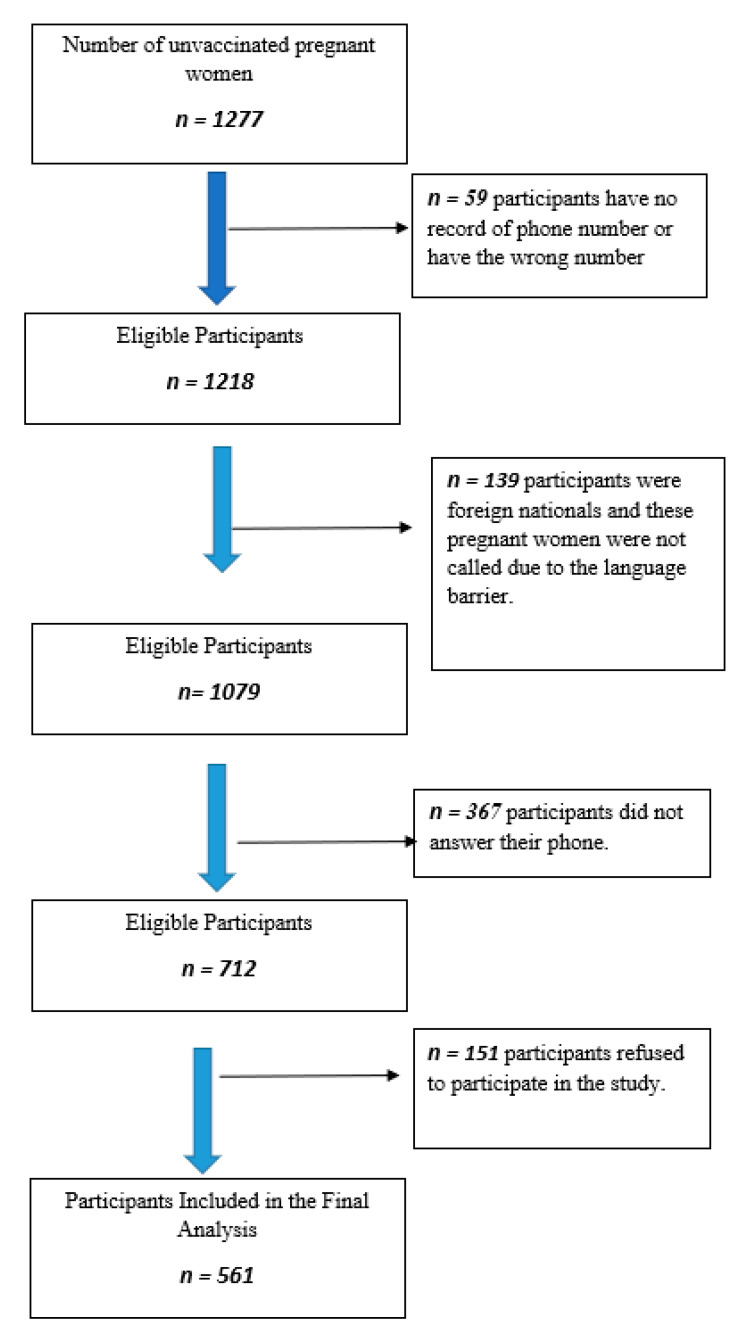
Flow chart on participants’ inclusion.

**Figure 2 vaccines-11-00132-f002:**
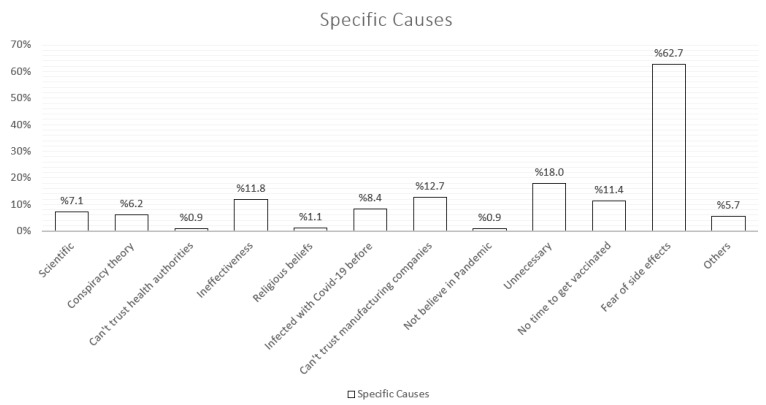
Identification of specific causes of vaccination refusal in unvaccinated pregnant women.

**Table 1 vaccines-11-00132-t001:** Comparison of the sociodemographic characteristics of the participants according to the scores they received from the vaccine hesitancy scale.

		Sample Size, *n* (%)	Vaccine Hesitancy Scale Scores		*p* Value *
			Mean	Sd ***	
Education	**Illiterate**	**30 (5.3)**	33.43	7.36	**0.005 ***
	Primary Education	305 (54.4)	32.17	6.01	
	High School	144 (25.7)	33.60	6.39	
	University	82 (14.6)	33.96	5.91	
Chronic Diseases	Yes	51 (9.1)	31.04	4.80	**0.027 ****
	No	510 (90.9)	33.05	6.30	
Income Status	Below 4682 TL	300 (53.5)	31.91	5.92	**<0.001 ****
	Above 4682 TL	261 (46.5)	33.96	6.34	
Pandemic effect on income	Yes	416 (74.2)	32.52	6.20	**0.013 ****
	No	145 (25.8)	33.86	6.12	
Occupation	Employed/Self-Employed	61 (10.9)	33.51	5.37	0.225 **
	Unemployed	500 (89.1)	32.79	6.30	
Infected with COVID-19	Yes	286 (51.0)	32.96	5.73	0.361 **
	No	275 (49.0)	32.77	6.67	
Hospitalization for COVID-19	Yes	11 (3.8)	32.91	7.44	0.524 **
	No	271 (96.2)	33.00	5.69	
Perceived risk of being infected with COVID-19	No risk	114 (20.3)	33.80	6.20	0.433 *
	Little risk	214 (38.1)	32.92	5.30	
	Moderate risk	200 (35.7)	32.27	6.25	
	High risk	32 (5.7)	33.09	6.30	
	Very high risk	1 (0.2)	29.00	7.85	
Anyone advised getting COVID-19 vaccine	Yes	331 (59.0)	32.45	6.38	**0.015 ****
	No	230 (41.0)	33.47	5.89	
Previous vaccination status with other vaccines	Yes	534 (95.2)	32.81	6.22	0.281 **
	No	27 (4.8)	34.00	5.90	
Tetanus vaccination status	Yes	470 (83.8)	31.93	5.50	**<0.001 ****
	No	91 (16.2)	37.69	7.32	

Bold values are significant with *p* < 0.05. CI− and CI+ are the lower and upper bonds of the 95% confidence interval. * Kruskal–Wallis Test ** Mann–Whitney U Test *** Standard Deviation.

**Table 2 vaccines-11-00132-t002:** Distribution of participants according to whether or not they received advice about getting vaccinated.

		*n*	%
The family doctor advised not to get vaccinated	Yes	4	0.7
	No	557	99.3
Friends/Neighbours advised not to get vaccinated	Yes	57	10.2
	No	504	89.8
Specialist advised not to get vaccinated	Yes	2	0.4
	No	559	99.6
Husband/Relatives advised not to get vaccinated	Yes	64	11.4
	No	497	88.6
Others advised not to get vaccinated	Yes	12	2.1
	No	549	97.9
The family doctor advised getting vaccinated	Yes	135	24.1
	No	426	75.9
Friends/Neighbours advised getting vaccinated	Yes	49	8.7
	No	512	91.3
Others advised advised getting vaccinated	Yes	6	1.1
	No	555	98.9
Husband/Relatives advised getting vaccinated	Yes	164	29.2
	No	397	70.8
Specialist advised getting vaccinated	Yes	104	18.5
	No	457	81.5

**Table 3 vaccines-11-00132-t003:** Multivariate analysis of variables *.

	*p* Value	OR	95% CI.for OR	
			Lower	Upper
Education	**0.026**			
Illiterate	0.892	1.064	0.436	2.597
Primary Education	**0.014**	0.520	0.308	0.878
High School	0.387	0.777	0.440	1.375
Chronic Disease (yes)	0.243	0.683	0.360	1.296
Pandemic Effect On Income (yes)	0.316	0.813	0.542	1.219
Anyone Advised Getting COVID-19 Vaccine (yes)	0.060	1.409	0.985	2.016
Tetanus Vaccination Status (no)	**<0.001**	3.681	2.237	6.056
Income Statue (Above 4682 TL)	**0.014**	1.585	1.098	2.287
Constant	0.90	0.803		

* Education status, chronic disease, income, the effect of the pandemic on income, whether someone recommends the COVID-19 vaccine, and the status of having tetanus vaccine included in the analysis. Bold values are significant with *p* < 0.05. CI− and CI+ are the lower and upper bonds of the 95% confidence interval.

## Data Availability

The data presented in this study are available on request from the corresponding author.

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
