# Peer review of "COVID-19 Vaccine Hesitancy and Related Factors among Unvaccinated Pregnant Women during the Pandemic Period in Turkey"

_vaccines, 2023, doi:10.3390/vaccines11010132_

Round 1

Reviewer 1 Report

Estimated Authors,

I've read with great interest the present study on the determinants of COVID-19 vaccination in pregnant women from Türkiye. A total of 561 subjects were included in the analysis, and "62.7% stated that they were afraid of the possible side effects of the vaccine and therefore did not want to be vaccinated against Covid-19". Authors have performed a series of analyses on the main effectors of this attitude, and primary education achieved and tetanus vaccination status (i.e. a proxy of vaccine attitude) were characterized as the main ones.

This paper is, from my point of view, of some interest and certainly consistent with the aims of Vaccines, but some improvements are required before its eventual acceptance.

More precisely:

Main issues: Tables 2+3 are unclear in their content. Authors provide a descriptive reporting, but they also report a p value for differences, whose reference is not properly stated, neither in the table or its caption. A more appropriate way for reporting this data would be reporting the overall score assessed and the individual comparisons, or the categorical comparison (i.e. total figures, and by being favorable towards vaccines vs. not being favorable). Please double check and eventually fix it in a more appropriate way, possibly consistent with the dichotomous comparison from multivariable analyses in table5.

Minor issues:

- Figure 1 should be fixed, as the pagination has impaired its readability;

- please fix comma notation of decimal figures as dot notation in main tables (i.e. 0.05 vs. 0,05)

- Captions and labels of the tables should be expanded in order to be more informative of their content

- economic values are reported in Turkish lira, but it could be unclear to the international reader; please report as high vs. low income, explaining in the main text how this cut off was implemented;

Author Response

Response to Reviewer 1 Comments

Point 1: Tables 2+3 are unclear in their content. Authors provide a descriptive reporting, but they also report a p value for differences, whose reference is not properly stated, neither in the table or its caption. A more appropriate way for reporting this data would be reporting the overall score assessed and the individual comparisons, or the categorical comparison (i.e. total figures, and by being favorable towards vaccines vs. not being favorable). Please double check and eventually fix it in a more appropriate way, possibly consistent with the dichotomous comparison from multivariable analyses in table5.

Response 1: Thank you very much for your valuable comment and we apologize for this detail that we missed. Tables 2 and 3 have been revised. Table 2 and 3 will appear as tables 1 and 2 in the manuscript, due to the fact that table 1 has been transformed into shape. Since the scale we used did not have a cut-off point, we conducted our analysis on the average score and our logistic regression analysis in such a way that those below the average had no hesitation about vaccination and those above the average had hesitation. And we edited the title and description in tables 1 and 2. You can see the corrections in the main manuscript that we uploaded.

Point 2: Figure 1 should be fixed, as the pagination has impaired its readability;

Response 2: Thank you for your suggestion. As you suggest, we fixed Figure 1, hope it seems better and readable now.

Point 3: Please fix comma notation of decimal figures as dot notation in main tables (i.e. 0.05 vs. 0,05)

Response 3: Thank you! We regret to leave it as we use it in Turkish. All have been fixed.

Point 4: Captions and labels of the tables should be expanded in order to be more informative of their content.

Response 4: Thank you for your suggestions. We revised the tables and expanded to be more informative.

Point 5: Economic values are reported in Turkish lira, but it could be unclear to the international reader; please report as high vs. low income, explaining in the main text how this cut off was implemented.

Response 5: Thank you for your comment. We added some information about it in methods section’s measuring tools part. You can see it in the main manuscript as corrections.

Reviewer 2 Report

The authors carried out a study on the vaccination hesitations of pregnant women who refused to be vaccinated for Covid-19 during the pandemic period. The manuscript is well-written and the method section is clear.

Figure 1 needs to be formatted correctly in the manuscript. Also, it is hard to digest table1. It would be better to replace it with a plot.

It would be helpful if the authors could discuss potential bias in the sampled population using the current sampling method.

Authors should make the data of the questionnaire publicly available.

Author Response

Response to Reviewer 2 Comments

Point 1: Figure 1 needs to be formatted correctly in the manuscript. Also, it is hard to digest table1. It would be better to replace it with a plot.

Response 1: Thank you for your comment. Figure 1 has been formatted correctly in the manuscript. You can see the corrections in the main manuscrpt that we uploaded. Also, Table 1 has been revised and change as Figure 2.

Point 2: It would be helpful if the authors could discuss potential bias in the sampled population using the current sampling method.

Response 2: Thank you for your suggestion. As you suggest, we have added what the selection bias resulting from the sample selection might be effective on, in the last paragraph of the discussion section, before mentioning the limitations.

Point 3: Authors should make the data of the questionnaire publicly available.

Response 3: Thank you! We will consider it and make our data of the questionnaire publicly available.

Round 2

Reviewer 1 Report

Estimated Authors,

first of all, thank you for the considerable efforts you paid in order to improve the overall quality of your study according to my previous recommendations.

In fact, I'm quite uncomfortable in both appreciating your work and recommending another round of revision before the full acceptance of this study, for the following reasons (but this time please note that we are only dealing with minor shortcomings)

1) Figure 1 and 2 have been improved, but some uncertainties still remain; maybe you did them through a program such as Libreoffice and then converted to a *.docx format? In case, I would suggest to preventively save the figures as images (e.g. jpeg / png) and then to embed them in the main file.

2) Table 1 has been clarified in that: we now clearly understand that the p value is associated with the continuous values reported in the penultimate column, but the table still conveys two very different kind of values: on the left half, % values, in the right one, the continues values. I would suggest to rework at least the caption of the table, through a more extensive explanation of its content.

3) the statement about the cut off values for income of participants improves the overall understanding of the paper but, by itself, it remains quite unclear: please, double check and edit it, particularly in the sentence I've marked through boldface font (i.e. "Income status of individuals was determined on the basis of 4682 TL, which is the hunger limit of a single person in Turkey at the time of data collection, and those whose income status above 4682 TL are good and below are interpreted as bad")

Author Response

Response to Reviewer 1 Comments

Point 1: Figure 1 and 2 have been improved, but some uncertainties still remain; maybe you did them through a program such as Libreoffice and then converted to a *.docx format? In case, I would suggest to preventively save the figures as images (e.g. jpeg / png) and then to embed them in the main file.

Response 1: Thank you for your suggestion. We have changed the figures by saving them as images and then embed in the main file. You can see in the main manuscript.

Point 2: Table 1 has been clarified in that: we now clearly understand that the p value is associated with the continuous values reported in the penultimate column, but the table still conveys two very different kind of values: on the left half, % values, in the right one, the continues values. I would suggest to rework at least the caption of the table, through a more extensive explanation of its content.

Response 2: Thank you so much. We are aware of its unclarity so we made some changes to it. We hope it looks better and is easy to understand now. We appreciate your suggestions.

Point 3: the statement about the cut off values for income of participants improves the overall understanding of the paper but, by itself, it remains quite unclear: please, double check and edit it, particularly in the sentence I've marked through boldface font (i.e. "Income status of individuals was determined on the basis of 4682 TL, which is the hunger limit of a single person in Turkey at the time of data collection, and those whose income status above 4682 TL are good and below are interpreted as bad")

Response 3: Thank you for your suggestion. We checked and edited it to make it more clear. You can see in the main manuscript by tracking changes.